# A Comparative Study on Hexavalent Chromium Adsorption onto Chitosan and Chitosan-Based Composites

**DOI:** 10.3390/polym13193427

**Published:** 2021-10-06

**Authors:** Rachid El Kaim Billah, Moonis Ali Khan, Young-Kwon Park, Amira AM, Hicham Majdoubi, Younesse Haddaji, Byong-Hun Jeon

**Affiliations:** 1Laboratory of Coordination and Analytical Chemistry, Department of Chemistry, Faculty of Sciences, University of Chouaib Doukkali, Avenue Jabran Khalil Jabran, El Jadida 24000, Morocco; rachidelkaimbillah@gmail.com; 2Chemistry Department, College of Science, King Saud University, Riyadh 11451, Saudi Arabia; 3School of Environmental Engineering, University of Seoul, Seoul 02504, Korea; catalica@uos.ac.kr; 4Laboratory of Analytical Chemistry and Physico-Chemistry of Materials, Department of Chemistry, Faculty of Sciences Ben M’Sik, University of Hassan II-Casablanca, Casablanca 20000, Morocco; Amamira6@gmail.com; 5Laboratory of Engineering and Materials, Department of Chemistry, Faculty of Sciences Ben M’Sik, University of Hassan II-Casablanca, Casablanca 21100, Morocco; hichammajdoubi.hm@gmail.com (H.M.); ys.haddaji@gmail.com (Y.H.); 6Department of Earth Resources and Environmental Engineering, Hanyang University, Seoul 04763, Korea; bhjeon@hanyang.ac.kr

**Keywords:** chitosan, silica, hydroxyaptite, chromium, adsorbent regeneration, water treatment

## Abstract

Chitosan (Cs)-based composites were developed by incorporating silica (Cs–Si), and both silica and hydroxyapatite (Cs–Si–Hap), comparatively tested to sequester hexavalent (Cr(VI)) ions from water. XRD and FT-IR data affirmed the formation of Cs–Si and Cs–Si–Hap composite. Morphological images exhibits homogeneous Cs–Si surface, decorated with SiO_2_ nanoparticles, while the Cs–Si–Hap surface was non-homogeneous with microstructures, having SiO_2_ and Hap nanoparticles. Thermal analysis data revealed excellent thermal stability of the developed composites. Significant influence of pH, adsorbent dose, contact time, temperature, and coexisting anions on Cr(VI) adsorption onto composites was observed. Maximum Cr(VI) uptakes on Cs and developed composites were observed at pH 3. The equilibration time for Cr(VI) adsorption on Cs–Si–Hap was 10 min, comparatively better than Cs and Cs–Si. The adsorption data was fitted to pseudo-second-order kinetic and Langmuir isotherm models with respective maximum monolayer adsorption capacities (q_m_) of 55.5, 64.4, and 212.8 mg/g for Cs, Cs–Si, and Cs–Si–Hap. Regeneration studies showed that composites could be used for three consecutive cycles without losing their adsorption potential.

## 1. Introduction

Water contamination is one of the most important global environmental concerns and is currently a subject of international attention [1]. Metal ions and organic xenobiotics are the two major classes of water contaminants [2]. Among metal-based water contaminants, chromium (Cr) is highly toxic. Thus, it has received considerable attention in recent years [1,3]. Chromium is emitted into aquatic environment through industrial activities such as steel manufacturing and electroplating processes. It is also used as an additive in making alloys, as a dye and mordant during dying process, and as a wood preservative. All these activities could be the potential carriers of Cr to aqueous environment [4].

In nature, Cr derivatives often exists in trivalent, Cr(III) and hexavalent, Cr(VI)) states, having different toxicities and mobility. Hexavalent chromium is a radioactive element, which could be carcinogenic. Even in trace, Cr(VI) poses potential threats to human health [5]. Thus, it is necessary to minimize its concentration in drinking and inland surface waters with maximum permissible limits set at 0.05 and 0.1 mg/L, respectively [6]. Numerous conventional treatment techniques including membrane filtration, precipitation [7], ion-exchange [8], and adsorption [9] have been engineered for the removal of Cr(VI) from water. Among them, membrane filtration and ion-exchange are the two most effective treatment technologies for the removal of Cr(VI) form water. However, these processes are expensive, thus, on economical ground they are not commercially applicable to remove Cr(VI) from water. Adsorption is a simple, versatile, and economical process for the removal of pollutants from wastewater. During this process, a solid phase (adsorbent) plays an important role to sequester pollutants (through physico-chemical forces) suspended in liquid phase (polluted water). Although the production and application of solid adsorbent materials has been one of our world’s fundamental goals since ancient times [10,11], the development of novel materials, porous materials specifically have recently gained considerable attention [12,13].

Chitosan (Cs) and apatite are porous materials with wide-range applications and, there, have been the subject of great interest in recent years. A study on natural and synthetic apatite has been produced to evaluate their heavy metals and organic substances adsorption capacities through the improvement of its textural properties such as specific surface area, porosity, and active sites [13]. Studies have shown an interest in fixing various types of molecules (polymer, acid, organic matter) on the apatite matrix, especially to hydroxyapatite (Hap) [14,15]. Hydroxyapatite is a biocompatible material with low water solubility, chemical stability, and excellent buffering capacity [16], while Cs is a chelating polycationic polymer, known for its non-toxicity, biocompatibility, and biodegradability. The presence of hydroxyl (-OH) and amine (-NH_2_) groups allows Cs to “chelate” with metal ions, making it an excellent adsorbent [16,17]. Silica (Si) is a highly porous amorphous material, generally used in blast furnaces remove sulphur-containing compounds from oil. Water molecules in variable amounts are chemically bound to Si gels surfaces, and are thus widely incorporated into organic and inorganic materials to consolidate and improve their structures [18,19]. Due to their porosity, higher specific surface area, and better mechanical and thermal stabilities, the Si gel-based adsorbents are commonly used for the removal of organic and inorganic pollutions from wastewater [20,21]. The surface-modified Si composites are cost-effective and easy to synthesize with minimal organic solvents requirement. These solid silicon oxide particles can be easily bonded to water-soluble polymers that have an affinity to chemically chelate heavy metal ions. Studies have shown the effectiveness of materials modification by Si gel for Cr ions adsorption. The presence of hydroxyl and silanol groups on the Si gel surface enhanced its affinity towards Cr ions adsorption [20,22,23].

Herein, Cs-based hybrid composites were developed by incorporating Si gel and Hap. First, Cs was extracted from shrimp shells through demineralization, deproteinization, and dealkylation steps. Thereafter, Cs was composited with Si gel to form chitosan-silica (Cs–Si) composite and with both Si gel and Hap to form chitosan-silica-hydroxyapatite (Cs–Si–Hap) composite. The hybrid Cs–Si–Hap composite was conjoined by covalent and ionic chemical bonds between the organic and inorganic phases. Covalent bonds (Si-O-C) were created as a result of a reaction between the carbinol groups of the polymer and the silanol groups of the Si lattice, while ionic bonds appear between the chitosan’s amino and silica’s silanol groups [24,25]. Weak interactions such as hydrogen bonds between the polymer’s amide or silica’s silanol groups can also be expected [26,27,28]. The developed composites were thoroughly characterized and their efficacy as adsorbents to sequester Cr(VI) ions from water were comparatively studied by applying kinetic, isotherm, and thermodynamic models. The effect of co-existing ions on Cr(VI) adsorption and regeneration potential of composite were also tested.

## 2. Experimental

### 2.1. Chemicals and Reagents

The chemicals and reagents used during the research were of analytical reagent (A.R) grade or as itemized. Potassium dichromate (K_2_Cr_2_O_7_, ≥99.5%), sodium hydroxide (NaOH, ≥99%), and 1, 5-diphenylcarbazide (C_13_H_14_N_4_O, ≥98%) were procured from Sigma-Aldrich, USA. Nitric acid (HNO_3_), hydrochloric acid (HCl), sodium chloride (NaCl), sulfuric acid (H_2_SO_4_), acetic acid (CH_3_COOH) and ethanol (C_2_H_5_OH) were purchased from Sigma-Aldrich, Germany. Deionized (D.I.) water from Milli-Q water purification system (Millipore, Burlington, MA, USA) was utilized during the study.

### 2.2. Preparation of Cs–Si–Hap Composite

#### 2.2.1. Preparation of Chitosan (Cs)

Crude chitin was extracted from shrimp shells (collected from local fish market) in two steps. During the first step, demineralization was performed at room temperature using 1N HCl solution (m/v ratio = 1:15). During the second step, deproteinization was carried out through refluxing at 90 °C under continuous stirring in 5% NaOH solution (m/v ratio = 1: 20) for 10 h. Thereafter, chitin was converted to Cs through chemical deacetylation process. During deacetylation, chitin was treated for 6 h with NaOH solution (48 % *w/v*) at 100 °C having solid–liquid ratio of 1: 20 to remove acetyl groups. The product was separated through vacuum filtration, washed several times with D.I water until neutral pH was reached, followed by drying at 50 °C for 48 h. 

#### 2.2.2. Preparation of Chitosan-Silica (Cs–Si) Composite

A total of 3.0 g of Cs was mixed in 90 mL CH_3_COOH (2% *v/v*) solution under continuous stirring for 1 h. The surface of the Si gel beads was activated by heating for 2 h at 110 °C. Thereafter, 1 g of Si was taken and immersed in 20 mL of D.I water to form slurry. The slurry was mixed with Cs mixture under continuous stirring for 2 h. Finally, the mixture was sonicated for 30 min in an ultrasonic bath. Thereafter, the Cs–Si composite was refrigerated for 24 h at 4 °C to undergo a complete cross-linking reaction [20]. Further, Cs–Si composite was washed to have a neutral pH and then oven dried for 48 h at 50 °C.

#### 2.2.3. Preparation of Chitosan–Silica–Hydroxyapatite (Cs–Si–Hap) Composite

A total of 6 g of Cs was suspended in 100 mL of CH_3_COOH (2% *v/v*) solution under continuous stirring for 1 h. The Cs solution was mixed with 2 g of Si slurry (prepared in Section 2.2.2) and 1 g of Hap. This mixture was continuously stirred for 24 h. Finally, the Cs–Si–Hap composite was washed with D.I water to neutral pH through vacuum filtration, and dried for 48 h at 50 °C. The synthesis of Cs–Si–Hap composite is schematically displayed in Figure 1.

### 2.3. Characterization

X-Ray diffraction (XRD; Bruker D8 Advance, Billerica, MA, USA) analyses of adsorbent samples were carried out at 45 kV/100 mA, using CuKα radiation with Ni filter. Fourier Transform Infra-Red (FTIR; Thermo-scientific spectrometer) was employed to take the IR spectra of adsorbent samples. The surface morphology of the samples were tested by scanning electron microscopy (SEM; Philips XL 30 ESEM (Acc spot Magn 20.00 kV). Thermogravimetric analysis (TG; Discovery TGA) was performed at a heating rate of 10 °C/min under nitrogen atmosphere. 

### 2.4. Adsorption, Coexisting Anions, and Regeneration Studies

Batch mode adsorption of Cr(VI) on Cs, Cs–Si, and Cs–Si–Hap was studied. A total of 50 mL of Cr(VI) solution at an initial concentration (C_o_): 100 mg/L and pH: 3 was individually treated with 0.05 g of the three adsorbents on a shaker incubator operated at 100 rpm for a contact time of 2 h at room temperature. At the equilibrium, the solid/solution phases were separated by filtration and the residual Cr(VI) concentration was measured quantitatively by UV-Vis spectrometer (UV-Vis 1200) at 540 nm (details in Appendix A [29,30] and the equilibrium adsorption capacity (*q*_e_, mg/g), and adsorption (%) were measured using the following expressions:(1)$$q_{e}\;=\;\left(C_{o}-C_{e}\right)\;\times \;\frac{V}{m}$$
(2)$$Adsorption\;\left(\%\right)\;=\;\frac{\left(C_{o}-C_{e}\right)}{C_{o}}\;\times 100$$
where, *C*_o_, and *C*_e_ (mg/L) are the initial and equilibrium adsorbate concentrations, *V* (L) is the volume of adsorbate solution, and *m* (g) is the mass of adsorbents.

The experimental parameters such as: initial pH of Cr(VI) solution was varied from 1 to 7, adsorbents mass (Cs, Cs–Si, and Cs–Si–Hap) was varied from 0.005 to 0.5 g, contact time was varied from 5 to 120 min (2 h), initial Cr(VI) concentration was varied from 25 to 400 mg/L, and temperature was varied from 298 to 333 K during the study.

The effect of coexisting anions on Cr(VI) adsorption onto Cs, Cs–Si, and Cs–Si–Hap were studied in presence of sulfate (SO_4_^2−^), nitrate (NO_3_^−^), and chloride (Cl^−^) at an C_o_: 100 mg/L. 

The reusability of Cs, Cs–Si, and Cs–Si–Hap for Cr(VI) adsorption was tested through regeneration studies. About 0.05 g of adsorbent samples were saturated for 1 h with 50 mL of 100 mg/L Cr(VI) solution. At equilibrium, the solid/solution phases were separated. The Cr(VI)-loaded Cs, Cs–Si, and Cs–Si–Hap were washed with D.I water to remove traces of adsorbed Cr(VI) from the surface, dried, and reused for five consecutive regeneration cycles. 

## 3. Results and Discussion

### 3.1. Characterization

#### 3.1.1. X-ray Diffraction (XRD) Analysis

The XRD patterns of Cs, Cs–Si, and Cs–Si–Hap composite are presented in Figure 2a. The diffraction patterns of Cs shows two peaks at 10.16° and 21.8° reflecting the polymorphic characteristics of Cs [17]. For the Cs–Si sample, it was observed that the presence of new lines located between 20° and 25° were attributed to Si, the peaks corresponding to Cs were still present. This confirmed that Si did not cap Cs surface during Cs–Si composite development. The XRD patterns of Cs–Si–Hap displayed new peaks at 2θ 26°, 31°, 32°, and 40° which correspond to the (211), (300), and (310) diffraction planes of Hap, respectively. It also agrees well with International Commission on Diffraction Data (ICDD No. 9-432). The Cs–Si peaks were still present, with a slight decrease in intensity and the notable dilation of the diffraction peaks. These changes in shape were observed after Cs and Si scaffolding, indicating that the Hap particles were uniformly dispersed in the Cs–Si polymer matrix.

#### 3.1.2. Fourier Transform Infrared (FT-IR) Analysis

FT-IR spectroscopy was applied to confirm the modification of Cs to Cs–Si, and Cs–Si–Hap composites, illustrated in Figure 2b. The Cs spectrum shows the absorption bands at 3283, 3363, 2900, 1665, 1595, 1543, 1375, 1148, 1076, and 1031 cm^−1^ were attributed to stretching vibration of hydroxyl (-OH) group, methylene (-CH_2_) group, amide I (C=O), amino (-NH_2_), and amide II (-NH), C-N (amide III), and C-O-C in Cs, respectively [18]. Upon modifying with Si (ie.Cs–Si) it was observed that some of these groups disappeared due to the embedding of Si in Cs matrix. However, the broad transmission band at 3430 cm^−1^ in Cs and Cs–Si samples and 3453 cm^−1^ in Cs–Si–Hap composite spectrum indicates the presence of water molecules in the samples and they were attributed to the stretching vibrations of -OH groups or of free H_2_O. The peaks near 500 cm^−1^, shown in Cs–Si spectrum were assigned to Si-O-Si which is out of the plane of bends and stretching modes. These results provide evidence for successful composite structure formation between Si and Cs. The vibrational peaks at 1104 and 806 cm^−1^ were assigned to the symmetric and asymmetric stretching modes of SiO_2_ groups [19]. The FT-IR spectrum of Cs–Si–Hap composite displayed strong bands at 1000–1100 and 500–600 cm^−1^, associated with stretching and bending vibrations of phosphate (PO_4_^3−^) group in Hap, respectively. The broadening of the band already present in the Cs–Si spectrum at 1050 cm^−1^ shows the presence of polymer and its interaction with the PO_4_^3−^ groups [20]. The bands at 1420–1485 and 875 cm^−1^ were due to carbonate (CO_3_^2−^) ions in Hap. The bands at 1550–1700 cm^−1^ were due to the modal superposition of the Hap’s -OH group and the chitosan amide I and amide II groups. The bands at 3600–3700 cm^−1^ and 2800–2950 cm^−1^ were attributed to the -OH groups present in Cs [21]. The stretching (vibration) bands of hydroxyapatite phosphate were at 1000–1100 cm^−1^ and the bending bands of phosphate were at 500–600 cm^−1^ [31]. The characteristic of CO_3_^2−^ were also visible between 1420 and 1485 cm^−1^ [32]. It should also be noted that after the reaction between Cs–Si and Hap, the absorption band of the -OH group around 1404 cm^−1^ of Cs–Si was found to be very weak, which affirmed the modification of the Cs–Si to Cs–Si–Hap composite.

#### 3.1.3. Scanning Electron Microscopy (SEM) Analysis

The morphology of Cs, Cs–Si, and Cs–Si–Hap composite were analyzed by SEM, illustrated in Figure 3a–c. The SEM image of Cs sample showed an uneven surface with some small cracks (Figure 3a), while the Cs–Si exhibits homogeneous distribution of SiO_2_ particles without affecting the actual morphology of Cs (Figure 3b) [33]. Moreover, certain SiO_2_ nanoparticles were incorporated into the Cs matrix, whereas other particles were dispersed on its surface. The Cs–Si–Hap composite exhibits a non-homogeneous surface with microstructures in which the Cs and Hap were mutually embedded along with the presence of Si particles while developing pores in the matrix (Figure 3c). 

#### 3.1.4. Thermal Analysis

The thermal behavior of Cs, Cs–Si, and Cs–Si–Hap composite was evaluated using thermogravimetric analysis (TGA) over the temperature range of 45–700 °C, and the results are presented in Figure 4a,b. The TGA curves indicate the occurrence of two-steps mass losses for Cs, Cs–Si, and Cs–Si–Hap (Figure 4a). The first-step, in the temperature range of 45–100 °C, was due to the removal of adsorbed water (dehydration) molecules [23]. The second mass loss was observed in the range of 200 to 400 °C, which was due to the decomposition of volatile organic matter for Cs and the thermal decomposition of the inorganic compound present in Cs–Si and Cs–Si–Hap composite [24]. Moreover, the DTG curves (Figure 4b) showed the maximum dissociation rate for Cs, Cs–Si, and Cs–Si–Hap occurred at 240, 250, and 280 °C, respectively. The results suggest excellent thermal stability of the composite materials to be applied as an adsorbent.

### 3.2. Parameters Affecting Cr(VI) Adsorption

#### 3.2.1. Effect of pH

The influence of pH on Cr(VI) uptake Cs, Cs–Si, and Cs–Si–Hap composite was investigated in pH range of 1.0 to 7.0, illustrated in Figure 5a. Increase in Cr(VI) adsorption was observed between pH 1 and 3. Thus, the maximum Cr(VI) uptakes on Cs, Cs–Si, and Cs–Si–Hap composite were 61.4, 78.6, and 99.8%, respectively, observed at pH 3. Interestingly, a similar adsorption performance was seen on Cs–Si and Cs–Si–Hap composite even at pH 4. Further increase in the solution pH from 4 to 7 results in decreased Cr(VI) adsorption. At pH 7, the Cr(VI) adsorption on Cs, Cs–Si, and Cs–Si–Hap was 18.6, 32.7, and 43.8%, respectively. Depending on the solution pH, the hexavalent chromium ions exist as oxyanions (HCrO_4_^−^, Cr_2_O_7_^2−^, and CrO_4_^2−^). In pH range 2–6, HCrO_4_^−^ and Cr_2_O_7_^2−^ species predominantly exist in aqueous solution [34,35]. The Cs contains polar functional groups such as amino groups (-NH_2_) in their molecular structure that can be involved in adsorption [36]. The surfaces of Cs–Si and Cs–Si–Hap composite were activated by the ions added to the Cs, i.e., Si, P, and Ca. These ions increase the ability to attract more Cr(VI) ions. At lower pH, the surface of the Cs based composites might become positively charged (-NH_3_^+^) due to protonation in the acidic medium. As a result, the adsorption sites increase. Therefore, at pH 3, an increase in Cr(VI) adsorption was observed due to the formation of more HCrO_4_^−^ ions based on the pH-diagram of Cr(VI) [37]. The excessive OH^−^ ions in the acidic pH neutralize the positive charges on the adsorbent surface so that the chromate ions (HCrO_4_^−^) can penetrate easily into the adsorbent pores. At pH > 3, the surface group can partially deprotonate, which inhibits the adsorption of Cr(VI). At this pH interval the dominant form was CrO_4_^2−^, which needs two adsorption sites, while hydrogen chromate (HCrO_4_^−^) needs only one active adsorption site based on its charge, so the decrease in Cr(VI) absorption was justified [38]. The adsorption of Cr(VI) on Cs, Cs–Si, and Cs–Si–Hap was found to be pH dependent with maximum adsorption occurred at pH 3, due to the electrostatic interactions between the adsorbent and adsorbate. The better removal percentage of Cr (VI) ions was achieved under acidic condition; consequently, pH 3 was chosen as optimal pH for the study. The point of zero charge (pH_PZC_) was determined by solid addition method [39] and the observed values for Cs, Cs–Si, and Cs–Si–Hap were 6.5, 6.2, and 6.6, respectively (Figure 5b).

#### 3.2.2. Effect of Adsorbent Dose and Contact Time

Figure 6a displayed the effect of variation in adsorbent dose (between 0.05 and 0.50 g) on Cr(VI) adsorption. In general, the increase in amount of adsorbent led to higher Cr(VI) uptake, since large number of unsaturated adsorption sites were available for binding Cr(VI) ions. About 92.9% of the Cr(VI) removal efficiency was achieved when using 0.05 g of Cs–Si–Hap composite, while no significant increase in Cr(VI) uptake was observed with increase in Cs–Si–Hap mass. On the other hand, the respective Cr(VI) uptakes on Cs and Cs–Si with similar mass (i.e., 0.05 g) were 54 and 65%, and reaches to 62 and 81% at 0.10 g dose, respectively. Further increase in both Cs and Cs–Si dose showed no significant increase in Cr(VI) removal efficiency.

Figure 6b showed the effect of contact time on Cr(VI) adsorption on Cs, Cs–Si, and Cs–Si–Hap composite. Two-step adsorption was observed. During the initial 10 min, the adsorption was rapid, with Cr(VI) uptake of 52% on Cs, 73% on Cs–Si, and 91% on Cs–Si–Hap. The equilibration time on Cs and Cs–Si were 30 and 20 min, respectively, while on Cs–Si–Hap composite, the equilibration time was 10 min. The initial rapid Cr(VI) uptake was due to external surface adsorption as large number of unsaturated adsorption sites were available. Slower and more stable Cr(VI) uptake was observed during conclusive step as adsorption was mainly governed by transporting Cr(VI) ions into the internal surface of the adsorbent (internal surface adsorption). The difference in equilibrium time between the adsorbents can be attributed to the surface modifications, which improves the tendency to rapidly remove Cr(VI) ions.

### 3.3. Adsorption Modeling

#### 3.3.1. Kinetic Modeling

The adsorption kinetic parameters for Cr(VI) adsorption on Cs, Cs–Si, and Cs–Si–Hap were evaluated by employing Lagergren’s pseudo-first-order [40] and Ho and Mckay’s pseudo-second-order kinetic models [41] (details in Appendix A). The linearized kinetic models plots are given in Appendix A and Table 1 presents kinetic parameters. The experimental and calculated adsorption capacities for Cr(VI) adsorption on Cs, Cs–Si, and Cs–Si–Hap by pseudo-first-order kinetic model were different. In addition, the correlation coefficient (R^2^) values were slightly lower than unity. These results confirm that pseudo-first-order kinetic model was not the best fitted model for Cr(VI) adsorption. Conversely, the experimental and calculated adsorption capacities for Cr(VI) adsorption on Cs, Cs–Si, and Cs–Si–Hap by pseudo-second-order kinetic model were nearer. In addition, the R^2^ values were slightly closer to unity. This affirms the applicability of pseudo-second-order model for Cr(VI) adsorption on Cs, Cs–Si, and Cs–Si–Hap. The fitting of pseudo-second-order kinetic model was supported well by nearer q_e,exp._ and q_e,2_ values. In addition, highest pseudo-second-order kinetic rate constant (k_2_) for Cr(VI) adsorption on Cs–Si–Hap affirmed comparatively better adsorption on Cs–Si–Hap. Additionally, this confirms that Cr(VI) adsorption was controlled by chemisorption.

#### 3.3.2. Isotherm Modeling

Langmuir [42] and Freundlich [43] isotherm models were applied during the adsorption studies (details in Appendix A and linearized isotherm plots are given in Appendix A). Langmuir model assumed the formation of a mono-layer on the adsorbent surface, existence of defined adsorption sites, and the surface is uniform with absence of interaction between the adsorbed molecules. The Freundlich model is based on an empirical equation reflecting a variation of energies with the amount adsorbed. This distribution of interacting energies is explained by the heterogeneity of the adsorption sites. Unlike Langmuir’s model, Freundlich’s equation does not predict an upper limit to the adsorption, which restricts its application to dilute mediums. However, this model admits the existence of interactions between the adsorbing molecules [44].

Table 2 presents isotherm modeling parameters. The Langmuir isotherm model provides a better fitting results depicted by higher R^2^ values, with maximum monolayer adsorption capacity (q_m_) values for Cr(VI) adsorption on Cs, Cs–Si, and Cs–Si–Hap were 55.51, 64.42, and 212.76 mg/g, respectively. The q_m_ increases with the substitution of Si and Hap into Cs matrix indicating an efficiency in the formulation of these composites as well as an affinity to adsorb Cr(VI) ions. The highest adsorption capacity was found on Cs–Si–Hap composite, while Cs–Si and Cs have relatively lower Cr(VI) uptake. Improved Cr(VI) uptake on Cs–Si–Hap composite could be explained by the synergic effect of amino reactive groups (present on Cs) as well as active binding sites provided by Hap containing phosphate groups. On the other hand, fitting of Langmuir model to adsorption data indicates that the adsorption of Cr(VI) ions on Cs, Cs–Si, and Cs–Si–Hap was through monolayer coverage and there was no interaction between adsorbed molecules. The q_m_ value of Cs–Si–Hap for Cr(VI) adsorption was comparatively better than other chitosan-based composites, reported in Table 3 [44,45,46,47,48,49,50]. In addition, Freundlich constant n values for Cs, Cs–Si, and Cs–Si–Hap were in a favorable adsorption range (between 0 and 10).

#### 3.3.3. Thermodynamic Parameters

The thermodynamic parameters for Cr (VI) ions on Cs, Cs–Si, and Cs–Si–Hap were evaluated in temperature range 298–333 K (details in Appendix A and Va not Hoff’s plots are given in Appendix A). Table 4 displayed thermodynamic studies data. Negative standard free energy change (∆*G*°) values for Cs, Cs–Si, and Cs–Si–Hap at varied temperatures affirmed the spontaneous nature of the Cr(VI) ion adsorption. The value of ∆*G*° becomes more negative with increasing temperature, indicating that the reaction tend towards favorability with increase in temperature. The positive standard enthalphy change (Δ*H*°) values indicate that the adsorption process was endothermic in nature [51]. The positive standard entropy change (Δ*S*°) values indicate the increase in randomness at the solid–liquid interface during the adsorption of Cr(VI) ions on Cs, Cs–Si, and Cs–Si–Hap.

### 3.4. Effect of Coexisting Anions on Cr(VI) Adsorption

Generally, in industrial effluents Cr(VI) ions often coexists with other ions which may affect the adsorption performance of an adsorbent. Therefore, it is essential to study the competitive influence of these coexisting ions on the removal of Cr(VI) by Cs, Cs–Si, and Cs–Si–Hap composite. Three co-existing anions viz. sulfate (SO_4_^2−^), nitrate (NO_3_^−^), and chloride (Cl^−^) at an C_o_: 100 mg/L were used to evaluate the adsorption capacity of Cr(VI) on Cs, Cs–Si, and Cs–Si–Hap and the results are displayed in Figure 7a. It can be seen that the three coexisting anions showed unfavorable effects on the removal of Cr(VI) on all the three adsorbents. Generally, the presence of coexisting ions, especially anions, could compete with Cr(VI) for occupying adsorption sites, leading to a decrease in Cr(VI) adsorption, especially SO_4_^2−^, which competes with HCrO_4_^−^ with two exchange ions, while Cl^−^ and NO_3_^−^ are weaker oxidizing agents than HCrO_4_^−^.

### 3.5. Regeneration Studies

Reusability is an important parameter to be verified for testing the economical efficacy of an adsorbent. The possible reuse of adsorbents was studied for five consecutive cycles using 0.5 N NaOH solutions as an eluent. Initially, for three consecutive regeneration cycles, the Cr(VI) removal efficiency of Cs, Cs–Si, and Cs–Si–Hap was almost constant (Figure 7b). During forth cycle, 7 and 4.8% drop in Cr(VI) uptake was observed on Cs and Cs–Si, respectively, while Cr(VI) uptake was slightly dropped on Cs–Si–Hap. During fifth regeneration cycle, the adsorption of Cr(VI) decreased further by 3.3% on Cs, 6.7% on Cs–Si, and 7.1% on Cs–Si–Hap. Hence, these results indicate that Cs, Cs–Si, and Cs–Si–Hap can be successfully reused for three consecutive regeneration cycles without losing in uptake potential.

## 4. Conclusions

The incorporation of Si and Hap on the surface of Cs results in the formation of novel efficient composites for Cr (VI) removal. The synthesized composites showed high efficiency to bind Cr (VI) ions under acidic pH conditions due to electrostatic interactions and chelation effects. However, the Cs–Si–Hap composite showed a better adsorption performance of Cr(VI) than Cs and Cs–Si, under the similar experimental conditions. The adsorption of Cr (VI) on the developed composites was rapid and efficient. The Langmuir isotherm model could be applied to describe the adsorption of Cr (VI) ions, on the composites, with maximum capacities of 55.51, 64.42, and 212.76 mg/g, for Cs, Cs–Si and Cs–Si–Hap, respectively. Finally, the experiments carried out have shown that the newly developed composites can be effectively reused for three consecutive regeneration cycles. Furthermore, the coexisting ions have a little hindrance on the Cr (VI) uptake.

## Figures and Tables

**Figure 1 polymers-13-03427-f001:**
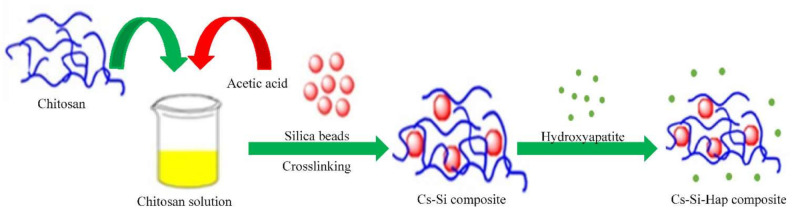
Schematic representation of Cs–Si–Hap composite synthesis.

**Figure 2 polymers-13-03427-f002:**
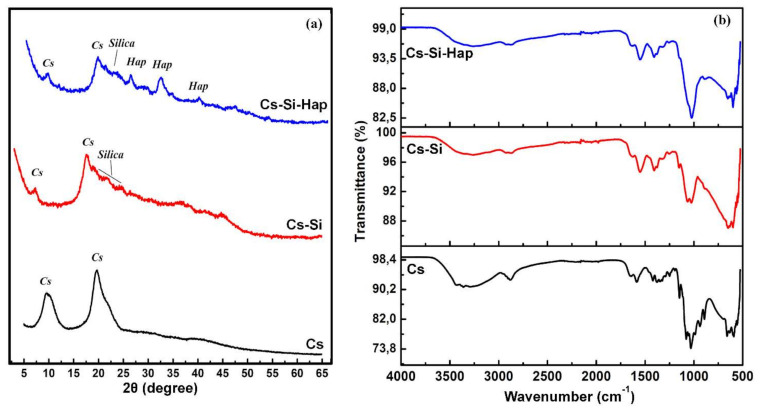
XRD patterns (**a**), and FT-IR spectra (**b**) of Cs- and Cs-based composites.

**Figure 3 polymers-13-03427-f003:**
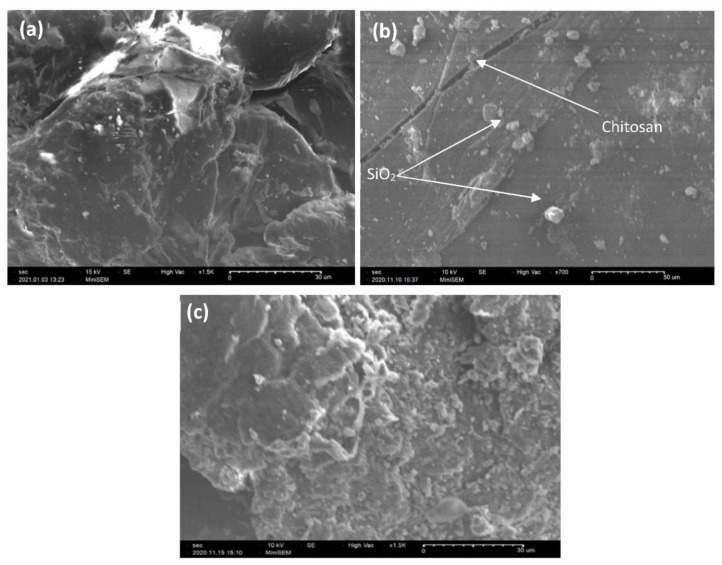
SEM images of Cs (**a**), Cs–Si (**b**), and Cs–Si–Hap (**c**).

**Figure 4 polymers-13-03427-f004:**
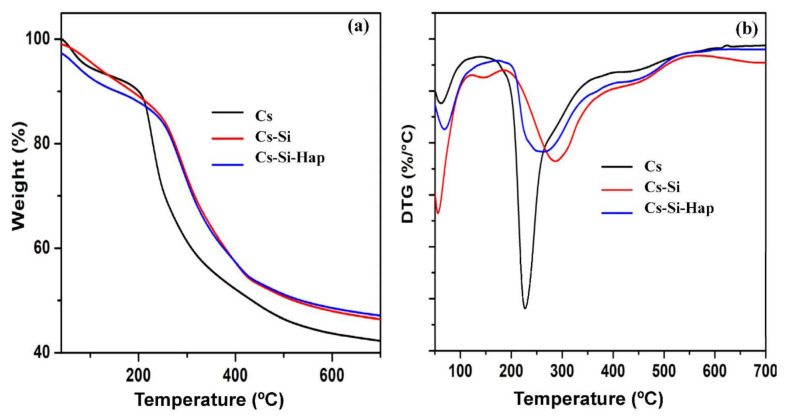
TGA (**a**), and DTG (**b**) analysis plots of Cs and Cs based composites.

**Figure 5 polymers-13-03427-f005:**
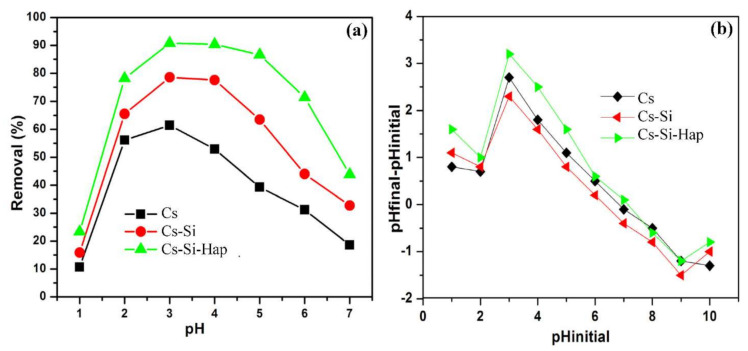
pH studies plots for Cr(VI) adsorption on Cs and Cs based composites (**a**), and point of zero charge (pH_PZC_) plots of Cs and Cs based composites (**b**).

**Figure 6 polymers-13-03427-f006:**
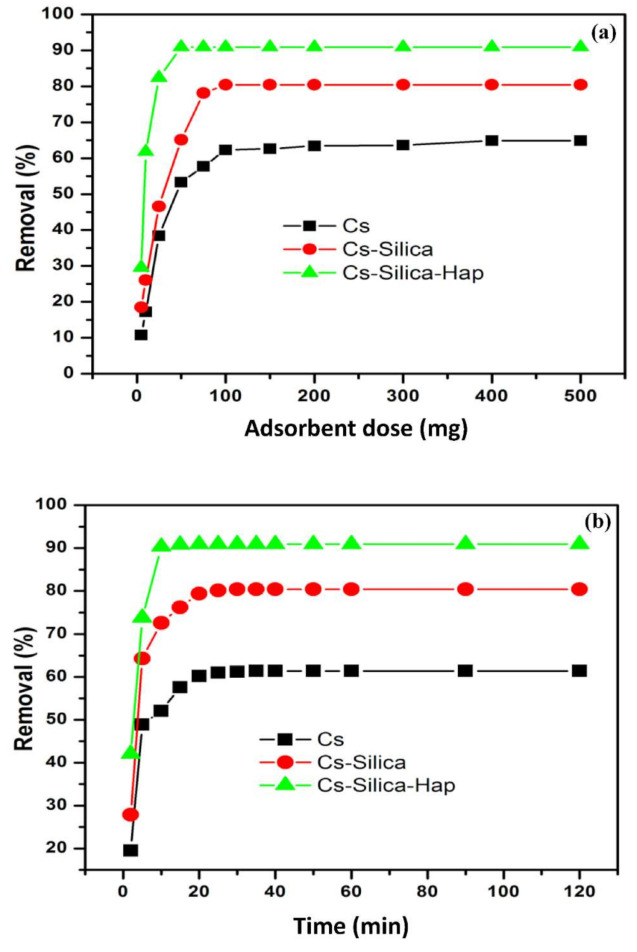
Adsorbent dose (**a**), and contact time (**b**) plots for Cr(VI) adsorption on Cs and Cs based composites.

**Figure 7 polymers-13-03427-f007:**
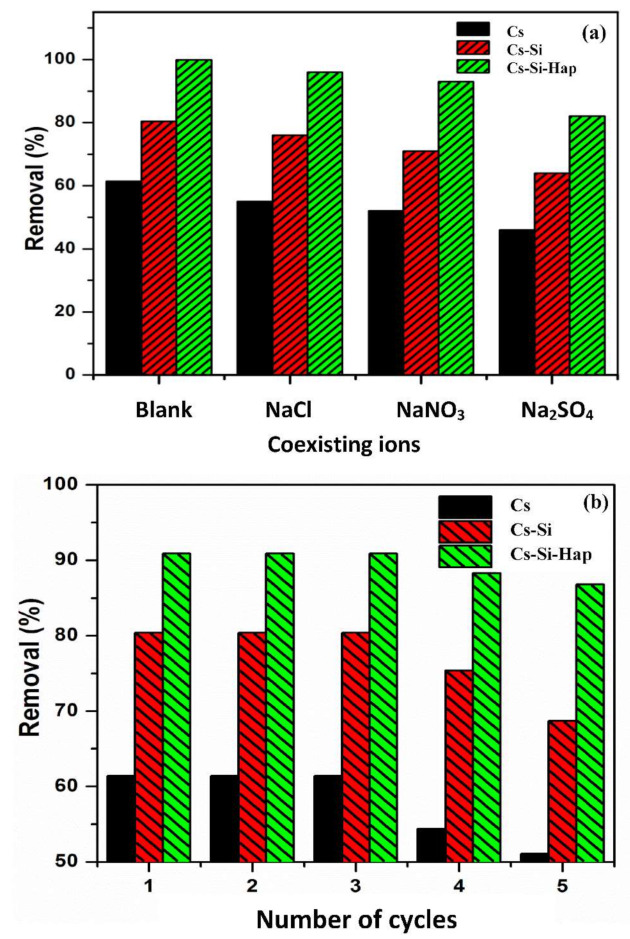
Effect of co-existing ions on Cr(VI) adsorption onto Cs and Cs based composites (**a**), and regeneration studies plot for Cr(VI) adsorption on Cs and Cs based composites (**b**).

**Table 1 polymers-13-03427-t001:** Kinetic models parameters for Cr(VI) adsorption.

Adsorbent	q_e,exp._ (mg/g)	Kinetic Models					
		Pseudo-First-Order			Pseudo-Second-Order		
		q_e,1_ (mg/g)	k_1_ (1/min)	R^2^	q_e,2_ (mg/g)	k_2_ (g/mg-min)	R^2^
Cs	31.35	24.89	0.070	0.974	30.70	0.020	0.998
Cs–Si	40.82	34.02	0.080	0.981	40.20	0.030	0.999
Cs–Si–Hap	91.74	104.45	0.212	0.983	90.90	0.040	0.999

**Table 2 polymers-13-03427-t002:** Isotherms models parameters for Cr(VI) adsorption.

Adsorbent	Isotherm Models					
	Langmuir			Freundlich		
	q_m_ (mg/g)	K_L_ (L/mg)	R^2^	K_F_ (mg/g)(L/mg)^1/n^	n	R^2^
Cs	55.51	0.080	0.991	7.91	2.88	0.971
Cs–Si	64.42	0.011	0.998	29.44	2.74	0.985
Cs–Si–Hap	212.76	0.230	0.999	31.86	3.71	0.989

**Table 3 polymers-13-03427-t003:** Comparison of q_m_ values for Cr(VI) adsorption on chitosan-based composites.

Adsorbent	Experimental Conditions	q_m_, mg/g	Reference
Cs-g-PMMA/ Silica BNC	C_o_: 10–500 mg/L; pH: 4; T: 25 °C; t: 24 h; m: 0.1 g; V: 20 mL; Agitation speed: 100 rpm.	92.50	[44]
Cs/triethanolamine/ Cu (II) composite	C_o_: 50–300 mg/L; pH: 8; T: 40 °C; m: 0.1 g; V: 50 mL.	44.64	[45]
Cs/GO/montmorillonite composite	C_o_: 25–250 mg/L; pH: 2; t: 3 h; m: 0.05 g; V: 50 mL	87.03	[46]
Cs/montmorillonite	C_o_: 6–24 mg/L; pH: 5; T: 30 °C; t: 3 h; m: 0.015 g; V:25 mL	35.71	[47]
Cs/MnFe_2_O_4_	C_o_: 0.1–1 mg/L; pH: 3; T: 20 °C; t: 12 h; m: 0.008 g; V: 300 mL.	31.32	[48]
Cs/GO/EDTA composite	C_o_: 20–100 mg/L; pH: 2; T: 25°C; t: 1.5 h; m: 0.02 g; V: 25 mL.	86.17	[49]
Cross-linked Cs-bentonite composite	pH: 2; t: 4 h; T: 20 °C; C_o_: 100–300 mg/L; m: 0.1 g; V: 20 mL.	37.73	[50]
Cs–Si–Hap	pH: 3; t: 1 h; T: 25 °C; C_o_: 20–140 mg/L; Agitation speed: 100 rpm.	222	This study

**Table 4 polymers-13-03427-t004:** Thermodynamic parameters for Cr(VI) adsorption.

Adsorbent	Thermodynamic Parameters				
	∆*H*° (kJ/mol)	∆*S*° (J/mol-K)	∆*G*° (kJ/mol)		
			298 K	308 K	333 K
Cs	57.37	190.85	−0.75	−1.56	−3.24
Cs-S	58.79	202.35	−1.47	−3.78	−5.65
Cs-S-Hap	33.15	135.30	−5.72	−7.15	−8.42

## Data Availability

Not applicable.

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
