# Peer review of "A Comparative Study on Hexavalent Chromium Adsorption onto Chitosan and Chitosan-Based Composites"

_polymers, 2021, doi:10.3390/polym13193427_

Round 1

Reviewer 1 Report

This manuscript presents a very interesting study in the development of materials with attractive properties to remove contaminants from liquid effluents. 

Affiliation: Department

Keywords: to improve the reach across platforms searches through specific words, I suggest do not use words already used in the title.

  1. Introduction

Pay attention to the use connectives in scientific language. Page number 1, line 34 (because). Page number 2, line 63 (on the other hand). I suggest to improve the flow and readability of the text.

In the last paragraph, page number 2, line 88, improve the aim of the paper. The authors should include the summary of what was developed in the adsorption stage, in relation to the final application for the material. Also, they should to mention the mathematical models used for kinetics, isotherms and thermodynamics.

  1. Experimental

The authors should include details regarding the colorimetric identification Cr (VI).

The methodology for potential of zero charge analysis was not included.

The equations for the mathematical models presented in the supplementary material should be cited in this chapter.

  1. Results and discussion:                                                 Characterization - Due to its application as an adsorbent material, it would improve the work to perform analysis of the specific surface area and pore size of the material developed.

3.1.2. FT-IR - In the page number 5, lines 182 and 185 the range discussed (cm-1) is not present in Figure 2b. Reference information about the functional groups obtained in the page number 5, lines 194 and 195.

3.1.3. SEM - In the case of pore analysis, add a discussion about the pores observed in the material structure. Identify figure 3 (c).

3.1.4. TGA - In Figure 4, identify (a) and (b).

3.2.1. Effect of pH - The most common way to shown the Figure is pHfinal - pHinitial on the y axis and this way allows to show when the line crosses the axis (zero). Discuss the results obtained and how they influence the adsorption process.

3.2.2 Effect of adsorbent mass and contact time - In general, the term used to study the effect of the adsorbent amount in adsorption processes is "adsorbent dosage". I suggest improve the discussion in this section. Improve de discussion in page number 8, lines 274, 275 and 276. Make it clearer.

3.3.1. Kinetic modeling - Improve the discussion of Table 1, explore the model that showed the best fitted of experimental data and the meaning of the parameters.

3.3.2 Isotherm modeling - Improve the discussion of Table 3. I suggest adding an error like ARE to the kinetic and isotherm data.

3.3.3 Thermodynamic parameters - Reference discussion.

3.4. Effect of coexisting anions on Cr(VI) adsorption - Improve the discussion. Make the results of the Figure 7a clearer. Was a single solution used or were the anions analyzed separately with Cr(VI)? The results obtained were satisfactory?

Supplementary material: Make sure to check the legend symbol and to include all information in the software used to obtain the figures.

Reviewer 2 Report

In this study, the author reported an efficient removal of Cr (VI) by a comparative survey of hexavalent chromium adsorption onto chitosan and chitosan-based composites. The manuscript is acceptable if authors can be improved the manuscript according to the below comments:

  • Would you mind thinking of a more specific title?
  • The abstract must be focused on valuable quantitative data for general audiences, which pave the way for persuading the audiences to read the full text.
  • Keywords should not include the title's contents.
  • The introduction should be targeted at a single main message, and in this format, it is not acceptable for length, scope, and clarity. I strongly recommend for revise of this part according to the below suggestions:
    1. Based on this study, what is the principal value instead of current literature, which is not clear in the introduction? Please emphasize the importance of adsorbents and their wide-range application via a comprehensive literature review table in the introduction.
    2. Developing nanofabricated photocatalysts and sorbents for hexavalent Cr removal remains a challenging issue, which must be confirmed with the literature. Please augment this part with pieces of evidence proposed in the following article: International Journal of Biological Macromolecules 188 (2021) 950-973 https://www.sciencedirect.com/science/article/abs/pii/S0141813021016421.
  • Please note that the captions are desired to be enriched via info, which makes this figure independently understandable instead of relying on the context.
  • Give more up-to-date references with the exclusion of possible self-citations.

Round 2

Reviewer 1 Report

Thank the authors for answering most of my questions. The reviewed paper is much better now. However, for one allegation, I suggest attention.

Please check the figures in supplementary material and include all informations (captions) *in the software* used to obtain them.

Reviewer 2 Report

accept
